# Marksman Backdoor: Backdoor Attacks with Arbitrary Target Class

**Khoa D. Doan**[1], **Yingjie Lao**[2], **Ping Li**[3]

[1]College of Engineering and Computer Science, VinUniversity
[2]Electrical and Computer Engineering, Clemson University, USA
[3]LinkedIn Ads, 700 Bellevue Way NE, Bellevue, WA 98004, USA
`khoa.dd@vinuni.edu.vn, ylao@clemson.edu, pinli@linkedin.com`

## Abstract

In recent years, machine learning models have been shown to be vulnerable to backdoor attacks. Under such attacks, an adversary embeds a stealthy backdoor into the trained model such that the compromised models will behave normally on clean inputs but will misclassify according to the adversary's control on maliciously constructed input with a trigger. While these existing attacks are very effective, the adversary's capability is limited: given an input, these attacks can only cause the model to misclassify toward a single pre-defined or target class. In contrast, this paper exploits a novel backdoor attack with a much more powerful payload, denoted as Marksman, where the adversary can arbitrarily choose which target class the model will misclassify given any input during inference. To achieve this goal, we propose to represent the trigger function as a class-conditional generative model and to inject the backdoor in a constrained optimization framework, where the trigger function learns to generate an optimal trigger pattern to attack any target class at will while simultaneously embedding this generative backdoor into the trained model. Given the learned trigger-generation function, during inference, the adversary can specify an arbitrary backdoor attack target class, and an appropriate trigger causing the model to classify toward this target class is created accordingly. We show empirically that the proposed framework achieves high attack performance (e.g., 100% attack success rates in several experiments) while preserving the clean-data performance in several benchmark datasets, including MNIST, CIFAR10, GTSRB, and TinyImageNet. The proposed Marksman backdoor attack can also easily bypass existing backdoor defenses that were originally designed against backdoor attacks with a single target class. Our work takes another significant step toward understanding the extensive risks of backdoor attacks in practice.

## 1   Introduction

Machine learning, especially deep neural networks (DNN), rapidly advances and transforms our daily lives in various fields and applications. Such intelligence is becoming prevalent and pervasive, embedded ubiquitously from centralized servers to fully distributed Internet-of-Things (IoT). Unfortunately, since well-trained models are now viewed as high-value assets that demand extensive computer resources, annotated data, and machine learning expertise, they are becoming increasingly attractive targets for cyberattacks [20, 51, 52]. Prior research has shown deep learning algorithms are vulnerable to a wide range of attacks, including adversarial examples [3, 31], poisoning attacks [33, 38, 17], backdoor attacks [30, 28, 15, 11], and privacy leakages [39, 14]. Among these, backdoor attacks expose the vulnerability in the model building supply chain that seeks to inject a stealthy backdoor into a model by poisoning the data or manipulating the training process [30, 28, 15]. Ideally, the model with injected backdoor should behave normally with clean inputs, but the input will be misclassified into the target class whenever the trigger is present.

36th Conference on Neural Information Processing Systems (NeurIPS 2022).

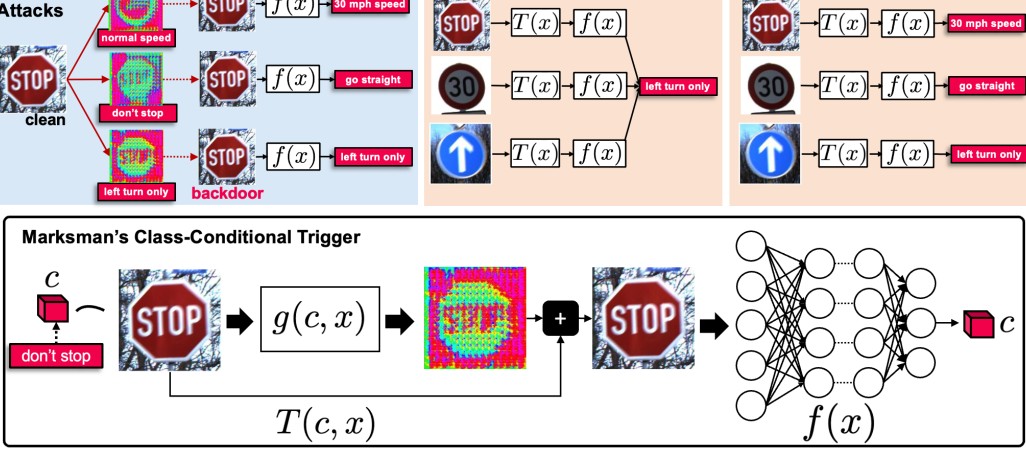

Figure 1: The payloads of Marksman and the existing backdoor attacks (top row). Marksman can attack an arbitrary target at will by generating a suitable trigger pattern via the class-conditional generative trigger function to cause the classifier to predict the chosen target (details in bottom row).

Past years have seen the development of backdoor attacks in various trigger forms, such as the patch-based in BadNets [15] and TrojanNN [28], blended and dynamic triggers [7, 37], and recently input-aware, invisible triggers [8, 34, 10, 9, 35]. In the fields of malware backdoor [4] and hardware Trojan [42, 48], these attacks are typically decomposed into two main components: the **trigger** that determines the activation mechanism and the **payload** that is used to control the modified malicious behavior. However, compared to the trigger mechanism, the payload of backdoor attacks on DNN is much less studied. The majority of the existing approaches consider either 1) all-to-one attacks where all the inputs with the trigger are mapped into one specific target class or 2) all-to-all attacks where the inputs from each true class will have different target labels [15]. As opposed to what the name of the all-to-all attacks indicates, such an attack is still only able to manipulate the inputs within one true class label into one target class. Nevertheless, these prior works are single-trigger and single-payload backdoor attacks with predefined target backdoor class(es). Even for the input-aware ones that generate the trigger based on the content of the input image to minimize the perceptual distinction, the target backdoor class is still predefined.

In this paper, we exploit the design of a backdoor attack with arbitrary target classes after injecting the backdoor into the model. Since we need to expand the payload capability of the backdoor attack, efficiently and effectively implementing such backdoor attacks is not trivial. One may argue that an adversary can repeatedly inject different trigger patterns for all the target classes to achieve such an adversarial objective. For instance, the adversary can use a specific trigger pattern for each target class and inject all the classes' trigger patterns into the model by using the patch-based backdoor strategy. Obviously, this method will lead to a much larger model perturbation than the single-trigger and single-payload attack. As the number of target classes increases, both the attack success rate (ASR) and the clean data accuracy will be significantly degraded.

To tackle these challenges, we follow the similar concept of invisible and input-aware backdoor attacks [8, 34, 10] that train a generative model during the backdoor injection, usually called trigger generator or trigger function, for generating triggers. In order to activate the backdoor, the adversary will feed the input image to the trigger generator/function, which will embed an input-specific trigger into the image. In these scenarios, the secret held by the adversary is the trigger function instead of fixed pixel patterns as in patch-based backdoor attacks. We efficiently incorporate the malicious functionalities that link to all the output classes into the trigger function to expand the payload. Consequently, as depicted in Figure 1, the adversary can arbitrarily choose the target class to misclassify given any input during inference, significantly enhancing the adversarial capability of the backdoor attack. The only works that we are aware of consider varying the payload are [49, 50], which yield different target classes by controlling the intensities of the same backdoor [49] and embedding multiple triggers into different channels (i.e., RGB channels) of an image [50], respectively. However, the possible numbers of target classes in these works are limited by 4 and 3, bounded by the

performance and the number of channels (i.e., 3 in RGB images). The work of DeepPayload [25] attempts to directly inject the malicious logic through reverse-engineering instead of training the backdoor into the model, which does not consider varying the target backdoor class as in this paper.

Our **contributions** are summarized below:

- We propose a new type of backdoor attack where the adversary can flexibly attack any target label during inference. This attack maliciously modifies the model by establishing a causal link between the trigger function and all output classes.

- We propose a class-condition generative trigger function that, given the target label, can generate an imperceptible trigger pattern to cause the model to predict the target label. We then propose a constrained optimization objective that can effectively and efficiently learn the trigger function and poison the model.

- Finally, we empirically demonstrate the effectiveness of the proposed method and its robustness against several representative defensive mechanisms. We show that the proposed method can achieve high attack success rates with any arbitrarily chosen target class while preserving the behavior of the model under normal conditions.

The rest of the paper is organized as follows. We review the background of DNN backdoor attacks in Section 2. The threat model is defined in Section 3. We present the details of the proposed methodology in Section 4, and evaluate the performance and compare to prior works in Section 5. Finally, Section 6 concludes this paper. We present more details about experimental settings and additional results in the supplementary material.

## 2   Background

### 2.1   Backdoor Attacks

Under image classification tasks, backdoor attacks on DNN seek to inject malicious behavior into the model that will associate a trigger to a target backdoor class [15, 28], which can also be interpreted as the payload as in malware backdoor and hardware Trojan. The injection of the backdoor are typically achieved by poisoning the training data [15, 28] or manipulating the training process or model parameters [19, 12]. An important performance requirement for the backdoor attack is its stealthiness, such that the existence of the backdoor in a model cannot be easily identified. Hence, a successful backdoor attack should preserve the normal functionality or inference accuracy for clean images (i.e., images without the trigger).

The designs of the trigger have been extensively studied in the literature, from the early obvious patch-based triggers [7, 28] to more invisible ones with utilization of blended [7], sinusoidal strips (SIG) [2], reflection (ReFool) [29], single-pixel [1], warping (WaNet) [35], discrete cosine transform (DCT) steganography [50], and adversarial example generation [41, 22]. As opposed to a universal trigger, several recent works have investigated input-aware backdoor attacks that minimize the visibility of the trigger by generating the trigger pattern based on the content of each input image [8, 34, 10, 9]. For instance, LIRA [10] trains a generative model as the trigger function for each image, while simultaneously injecting the backdoor into the model, which has been shown to be able to generate completely invisible triggers.

All of these attacks, under either all-to-one or all-to-all scenarios, can only manipulate the prediction of a given input image to one target class. While the works in [49, 50] considered a less narrow form for the payload, the number of possible target backdoor classes is still limited, i.e., 3 or 4. In contrast, this paper exploits a much stronger attack that is able to misclassify a given input image to any arbitrary target class.

### 2.2   Backdoor Defenses

Meanwhile, various backdoor defensive solutions have also been developed, aimed at either detecting [5, 44, 13] or mitigating [26, 45, 6, 36, 24] the attacks. Popular methods include Neural Cleanse [45] that detects the backdoor by searching for possible trigger patches, fine-pruning [26] that prunes the model to erase the backdoor, spectral signature [44] that detects outliers based on the latent representations, and STRIP [13] that uses perturbations to detect potential backdoor triggers.

Besides, input mitigation methods have also been studied, which seek to filter the images with triggers to avoid the activation of the backdoor [30, 24].

A successful backdoor attack has to be able to bypass the existing defenses. We evaluate our proposed Marksman backdoor against representative defensive solutions in our experiments.

## 3 Threat Model

Consistent with prior works on input-aware backdoor attacks that train a generative model for trigger generation [8, 34, 10, 11], we also consider the threat model where the adversary has full access to the model. Note that our setting is different from some backdoor attacks that only target poisoned data generation [15, 28]. During the training phase, the adversary attempts to inject the backdoor into a model. After that, the model will be delivered to victim users, who might employ the existing backdoor defensive measures to check the model. During the inference phase, the adversary is able to query the victim model with any inputs.

## 4 Proposed Methodology: Marksman Backdoor

### 4.1 Preliminaries

Consider the supervised learning setting where the goal is to learn a classifier $f_\theta : \mathcal{X} \to \mathcal{Y}$ that maps an input $x \in \mathcal{X}$ to a label $y \in \mathcal{Y}$. In empirical risk minimization (ERM), the parameters $\theta$ are learned using a training dataset $\mathcal{S} = \{(x_1, y_1), ..., (x_N, y_N)\}$ where $x_i \in \mathcal{X}$ and $y_i \in \mathcal{Y}$.

In a standard backdoor attack, a subset of $M$ ($M < N$) examples are first selected from $\mathcal{S}$ to create the poisoned subset $\mathcal{S}_p$. Each sample $(x, y)$ in this subset is transformed into a backdoor sample $(T(x), \eta(y))$, where $T : \mathcal{X} \to \mathcal{X}$ is the trigger function and $\eta$ is the target labeling function. The trigger function $T$ determines how a trigger pattern is placed on the input $x$ to create the backdoor input $T(x)$, while the target labeling function specifies how the classifier should predict in the presence of the backdoor input. The remaining samples in $\mathcal{S}$ comprise the clean subset $\mathcal{S}_c$, i.e., $\mathcal{S}_c = \mathcal{S} \setminus \mathcal{S}_p$.

Under ERM, we can alter the behavior of the classifier $f$ (i.e., inject the backdoor) by training $f$ with both the clean samples $\mathcal{S}_c$ and the backdoor samples $\mathcal{S}_p$, as follows:

$$\theta^* = \arg\min_\theta \sum_{(x,y) \in \mathcal{S}_c \cup \mathcal{S}_p} \mathcal{L}(f_\theta(x), y).$$

where $\mathcal{L}$ is the classification loss, e.g., cross-entropy loss. During inference, for a clean input $x$ and its true label $y$, the learned $f$ will behave as follows:

$$f(x) = y, \quad f(T(x)) = \eta(y)$$

### 4.2 Marksman's Payload: Arbitrary Attack Target Class

The training process described in the previous section essentially induces the payload, or the causal association between the trigger and the target label. As we discussed above, there are two common types of payload in the backdoor domain [15]: all-to-one and all-to-all. Under the all-to-one attack, all input with the trigger are predicted with a constant label, denoted as $\hat{\mathbf{y}}$, regardless of the original label $y$:

$$f(T(x)) = \hat{\mathbf{y}}, \quad \forall(x, y)$$

For the all-to-all attack, the input with the trigger is predicted with a label that depends on its true label $y$; for example, a commonly studied target function is

$$f(T(x)) = (y + 1) \bmod |\mathcal{Y}|, \quad \forall(x, y)$$

Note that, for both the all-to-one and all-to-all attacks, the attacker can only trigger one predefined target label. During inference, given an input, it is not possible to causally force $f$ to predict an arbitrary choice of a target label.

The goal of this work is to design a backdoor attack with a more flexible and powerful payload where the attacker can arbitrarily choose any target label during inference. Formally, we aim to alter the behavior of $f$ so that:

$$f(x) = y, \quad f(T(c, x)) = c, \quad \forall c \in \mathcal{Y} \text{ and } \forall (x, y) \tag{1}$$

Under this setting of Eq. (1), the trigger pattern, devised by the function $T$, is associated with the attack target $c$. This represents a more flexible payload design and a stronger backdoor attack because the adversary can freely choose to attack any target label during inference.

### 4.3 Learning the Marksman's Trigger Function

Modeling and eventually learning the trigger function to achieve the objective (i.e., injecting the backdoor into $f$) in Eq. (1) face some crucial challenges. First, the design of the trigger function should be systematic. Each causal association between an image and a specific target label is embedded in one unique trigger pattern. Thus, a large number of trigger patterns is needed to capture the multi-trigger and multi-payload requirement. Second, the injection of these trigger patterns into the model should preserve the clean-data performance while achieving a high attack success rate. On top of these, we also empirically observe that the straightforward baseline approach, which repeatedly injects these trigger patterns using the existing backdoor-attack methods such as the patch-based attack [15], fails to achieve this adversarial objective: either the clean-data performance noticeably drops below an acceptable threshold, or the attack success rate is not satisfactory.

To tackle these challenges, we first propose representing the process of creating the trigger pattern with a class-conditional generative model. The trigger function $T_\xi : \mathcal{Y} \times \mathcal{X} \to \mathcal{X}$ learns to generate a trigger pattern that is conditioned on the attack's target class and imperceptibly blended into the input image. The input to the generative model is a label $c$ and an image $x$, and its output is a backdoor input $T(c, x)$ with a trigger pattern that can activate the backdoor to attack the target label $c$. Formally, we can model $T$ using a class-conditional noise (or trigger pattern) generator $g$ as follows:

$$T(c, x) = x + g(c, x), \quad ||g(c, x)||_\infty \le \epsilon \tag{2}$$

where the $\infty$-norm constraint is used to ensure a small perturbation (within an $L_\infty$-ball with radius $\epsilon$) of the clean image for optimal visual stealthiness. The proposed class-conditional generative trigger gives the adversary a flexible control for systematically creating the trigger for any target class, as illustrated in Figure 1.

Given the class-conditional generative trigger function $T(c, x)$, we need to learn its parameters $\xi$ and to poison the classifier $f$. Recall that the objective is to preserve the clean-data performance while successfully performing the backdoor attack. Inspired by recent works in learning input-aware attacks [8, 34, 10, 11], we propose to jointly learn $T(c, x)$ and poison $f$ via the following constrained optimization objective:

$$\min_\theta \sum_{(x,y) \in \mathcal{S}_c} \mathcal{L}(f_\theta(x), y) + \alpha \sum_{\substack{(x,y) \in \mathcal{S}_p \\ c \ne y}} \mathcal{L}(f_\theta(T_{\xi^*(\theta)}(c, x)), c) \tag{3}$$

$$s.t. \quad \xi^* = \arg\min_\xi \sum_{(x,y) \in \mathcal{S}_p, c \ne y} \mathcal{L}(f_\theta(T_\xi(c, x)), c) - \beta ||g(c, x)||_2$$

In this problem, an optimal generative trigger function $T_{\xi^*}$ is associated with an optimally poisoned classifier. The poisoning process seeks the parameters $\theta$ of the classifier to minimize a linear combination of the clean accuracy and backdoor objectives. At the same time, the generative trigger function learns to optimally craft the backdoor images, conditioned on arbitrary target labels. To avoid trivial solutions (i.e., sparse and small trigger patterns), we also maximize the $L_2$-norm of the trigger pattern (the output of $g$) within the $L_\infty$-ball. This allows the generated trigger patterns to stretch over the entire input image. The hyperparameter $\alpha$ balances the mixing weights of the clean and backdoor objectives in training $f$ while the hyperparameter $\beta$ controls the effect of the penalty on sparser and smaller trigger patterns.

## 4.4 Marksman's Optimization

To solve the non-convex, constrained optimization in Eq. (3), we can alternately update $f$ and $T$ while fixing the other one. However, since our ultimate goal is to learn the poisoned $f$, constantly updating $T$ will cause the training process to converge very slowly. Inspired by some commonly accepted optimization tricks in reinforcement learning [32], we propose to update $T$ less frequently. During training, $f$ is updated at every alternating step, while $T$ is only updated once in several steps, e.g., after every training epoch. The backdoor samples to train $f$ at a step are generated using $T$ with parameter $\xi$ updated in the previous epoch. We observe that this approach effectively stabilizes the training process and allows $f$ to reach the optimal clean-data and attack performance in fewer training epochs. The details are presented in Algorithm 1.

---

**Algorithm 1** Marksman Optimization Algorithm

**Input:**
    (1) $S = \{(x_i, y_i), i = 1, ..., N\}$
    (2) training iterations $K$
    (3) iterations $k$ to update $T$
    (4) learning rate $\gamma_f$ and $\gamma_T$
    (5) batch size $b$, hyperparameters $\alpha$ and $\beta$

**Output:**
    learned parameters $\xi$ for $T$ and $\theta$ for $f$

1: Initialize $\theta$ and $\xi$, $\hat{\xi} \leftarrow \xi$, $j \leftarrow 0$
2: **repeat**
3:     Sample minibatch $\mathcal{S}_c = \{(x, y)\} \subset \mathcal{S}$
4:     Sample $S_p = \{(x, c) : c \neq y, (x, y) \in S\}$
5:     $\theta \leftarrow \theta - \gamma_f \nabla_\theta(\sum_{\mathcal{S}_c} \mathcal{L}(f_\theta(x), y) + \alpha \sum_{\mathcal{S}_p} \mathcal{L}(f_\theta(T_\xi(c, x)), c))$
6:     $\hat{\xi} \leftarrow \hat{\xi} - \gamma_T \nabla_{\hat{\xi}}(\sum_{\mathcal{S}_p} \mathcal{L}(f_\theta(T_{\hat{\xi}}(x)), c) - \beta||g(c, x)||_2)$
7:     $\xi \leftarrow \hat{\xi}$ if $j\%k = 0$, $j \leftarrow j + 1$
8: **until** $j = K$

---

## 5 Evaluation

### 5.1 Experimental Setup

We demonstrate the effectiveness of the proposed Marksman backdoor through comprehensive experiments on several widely-used datasets for backdoor attack study: **MNIST**, **CIFAR10**, **GTSRB**, and TinyImagenet (**T-IMNET**). We follow the previous works [41, 44, 5, 35] and consider various architectures for the classifier $f$: a CNN model [35] for MNIST, Pre-activation Resnet18 (**PreActResnet18**) [16] for CIFAR10 and GTSRB, and **Resnet18** [16] for T-IMNET.

As mentioned in previous sections, to the best of our knowledge, Marksman is the first work that studies multi-trigger and multi-payload backdoor with the capability of misclassifying an input to any target class. For comparison, we design three baseline approaches, **PatchMT**, **RefoolMT** and **WaNetMT**, by injecting different trigger patterns repeatedly for different target classes using recently studied trigger injection mechanisms. PatchMT is a multi-trigger and multi-payload extension of BadNets [15] where we use a different patch-based trigger pattern for a target class. Similarly, ReFoolMT is a multi-trigger and multi-payload extension of ReFool [29] where we use different reflection images for different target classes. Finally, WaNetMT is a multi-trigger and multi-payload extension of WaNet [35] where we randomly sample different combinations of the control-grid size $k$ (between 4 and 8) and the warping strength $s$ (between 0.5 and 1.0) for different target classes.

**Hyperparameters:** For each method, we train the classifiers using the momentum SGD optimizer (initial learning rate of 0.01, and after every 100 epochs, the learning rate is decayed by a factor of 0.1). For Marksman, we train $f$ and $T$ alternately but update $T$ slowly after every epoch. To achieve a high-degree visual stealthiness of Marksman, we select $\epsilon$ as small as 0.05 for all datasets. We perform grid searches to select the best $\alpha$ and $\beta$ using the CIFAR10 dataset and use these values for all the experiments. We also observe that a value of $\alpha$ closer to $1.0$ (specifically, around 0.8) achieves the optimal poisoning objective (i.e., highest clean-data accuracy and best ASR). More details of the experimental setup can be found in the supplementary material.

### 5.2 Effectiveness of Marksman

This section presents the attack success rates of Marksman and PatchMT. To evaluate the performance in the multi-trigger multi-payload scenario, for each test sample $(x, y)$, we enumerate all possible target labels $c$ other than the true label $y$ and use the compared attack method to activate the backdoor associated with these target labels. For each clean image, our attack only needs to feed it to the trigger function with a specified target class for generating the image with the trigger. The attack is considered successful for each input $x$ and a target label $c$ when the $f$ correctly predicts $c$.

Table 1: Clean and attack performance with 50% poisoning rate. Red values represent the performance drop w.r.t the original benign classifier.

| Dataset | PatchMT | | RefoolMT | | WaNetMT | | Marksman | |
|---|---|---|---|---|---|---|---|---|
| | Clean | Attack | Clean | Attack | Clean | Attack | Clean | Attack |
| MNIST | 0.967/*0.022* | 0.996 | 0.942/*0.047* | 0.893 | 0.970/*0.019* | 0.909 | 0.988/*0.001* | 1.000 |
| CIFAR10 | 0.882/*0.058* | 0.990 | 0.910/*0.030* | 0.984 | 0.920/*0.020* | 0.999 | 0.941/*0.007* | 1.000 |
| GTSRB | 0.943/*0.051* | 0.993 | 0.909/*0.085* | 0.977 | 0.962/*0.032* | 0.999 | 0.986/*0.001* | 0.999 |
| T-IMNET | 0.527/*0.052* | 0.951 | 0.429/*0.150* | 0.843 | 0.548/*0.031* | 0.999 | 0.577/*0.002* | 0.999 |

Table 2: Clean and attack performance with 10% poisoning rate. Red values represent the relative performance drop w.r.t the original benign classifier.

| Dataset | PatchMT | | RefoolMT | | WaNetMT | | Marksman | |
|---|---|---|---|---|---|---|---|---|
| | Clean | Attack | Clean | Attack | Clean | Attack | Clean | Attack |
| MNIST | 0.975/*0.014* | 0.298 | 0.977/*0.012* | 0.341 | 0.969/*0.020* | 0.784 | 0.983/*0.006* | 1.000 |
| CIFAR10 | 0.933/*0.007* | 0.487 | 0.934/*0.006* | 0.730 | 0.894/*0.046* | 0.308 | 0.943/*0.005* | 1.000 |
| GTSRB | 0.958/*0.031* | 0.376 | 0.951/*0.043* | 0.802 | 0.953/*0.041* | 0.012 | 0.979/*0.010* | 0.997 |
| T-IMNET | 0.577/*0.002* | 0.003 | 0.575/*0.004* | 0.137 | 0.562/*0.017* | 0.376 | 0.575/*0.004* | 0.999 |

### 5.2.1 Attack Performance

The clean-data accuracy (**Clean**) and attack success rates (**Attack**) of Marksman and the baselines are presented in Table 1 and Table 2 when poisoning 50% and 10% of the clean samples, respectively. As we can observe from these tables, both PatchMT's and ReFoolMT's clean-data performances drop significantly at a higher poisoning rate (50%), while their attack performances are very low (below 50%) at a lower poisoning rate (10%). As expected, when the number of labels in the classification task is higher (e.g., 200 for T-IMNET), the attack's success rates of PatchMT and ReFoolMT also decrease (e.g., reaching the performance, ≈ 0.003, of a random attack on 10% poisoning rate). For WaNetMT, the clean-data accuracy is better at the 50% poisoning rate, but there is still a non-trivial performance degradation, compared to Marksman. At the lower poisoning rate of 10%, which is a more reasonable rate, the attack performance is significantly worse than Marksman.

In comparison, Marksman is highly effective: 100% attack success rates on almost all datasets. Marksman can achieve such high ASRs without sacrificing the clean-data performance (only a trivial drop in performance). Since the objective of our method is to have the capability of misclassifying a given input to any target class during inference, we also report the attack success rate for each class of both Marksman and the baselines in Table 3. It can be seen that the Marksman achieves excellent performance for any arbitrary target classes. The success of Marksman confirms the threat of such an attack with a significantly more flexible and powerful payload than the previous studies in the backdoor domain and encourages future research on the corresponding defensive approaches.

Table 3: Attack success rate for each target class with 10% poisoning rate.

| **MNIST** | 1 | 2 | 3 | 4 | 5 | 6 | 7 | 8 | 9 | 10 |
|---|---|---|---|---|---|---|---|---|---|---|
| PatchMT | 0.373 | 0.209 | 0.162 | 0.267 | 0.288 | 0.390 | 0.149 | 0.368 | 0.172 | 0.621 |
| ReFoolMT | 0.720 | 0.230 | 0.954 | 0.006 | 0.050 | 0.131 | 0.420 | 0.882 | 0.031 | 0.009 |
| WaNetMT | 0.726 | 0.853 | 0.820 | 0.760 | 0.721 | 0.799 | 0.649 | 0.874 | 0.791 | 0.817 |
| Marksman | 0.997 | 0.998 | 1.000 | 1.000 | 0.999 | 1.000 | 1.000 | 1.000 | 0.998 | 0.998 |
| **CIFAR10** | 1 | 2 | 3 | 4 | 5 | 6 | 7 | 8 | 9 | 10 |
| PatchMT | 0.397 | 0.362 | 0.449 | 0.744 | 0.418 | 0.534 | 0.725 | 0.369 | 0.384 | 0.399 |
| ReFoolMT | 0.787 | 0.844 | 0.707 | 0.791 | 0.804 | 0.725 | 0.864 | 0.654 | 0.569 | 0.532 |
| WaNetMT | 0.290 | 0.330 | 0.316 | 0.428 | 0.324 | 0.391 | 0.241 | 0.398 | 0.242 | 0.354 |
| Marksman | 1.000 | 1.000 | 1.000 | 0.999 | 1.000 | 1.000 | 1.000 | 1.000 | 0.999 | 1.000 |

### 5.2.2 Attack Performance with Different Poisoning Rates

We can understand the challenges in adapting the existing backdoor attack methods for the multi-trigger multi-payload scenario and the effectiveness of Marksman by investigating the clean and backdoor performance of PatchMT and Marksman, respectively. Figure 2 shows the clean-data accuracy and ASR for different poisoning rates (i.e., $|\mathcal{S}_p|/(|\mathcal{S}_p| + |\mathcal{S}_c|)$) between 5% and 50%. We can observe that, for PatchMT, the classifier either performs noticeably worse on clean data with higher attack performance or preserves the clean-data performance with significantly lower ASR. In contrast, the performance of Marksman is consistent across different poisoning rates.

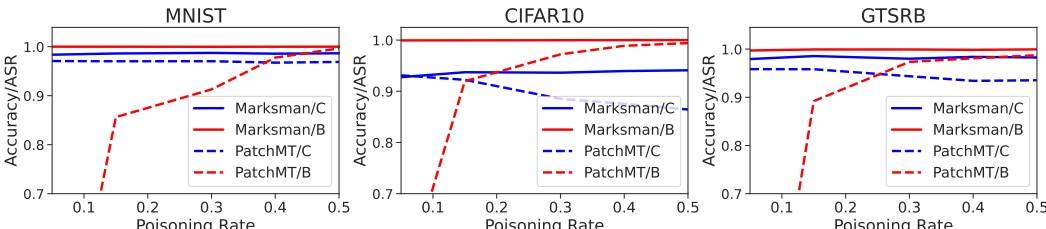

Figure 2: Clean (**/C**) and attack (**/B**) performance with different poisoning rates.

### 5.2.3 Performance of Trained Trigger Function on Other Models

While Marksman's threat model assumes that the adversary has full control of the training process, in this section, we demonstrate that the learned class-conditional trigger function can be used to attack other classifiers. In other words, Marksman can still be effective when the adversary has only access to the training data but does not control the training process or knows the internal structure of $f$. To perform the experiments, we freeze the parameters of the class-conditional trigger function $T$ after it is jointly learned with a classifier $f$, then generate poisoned samples to attack another classifier $\hat{f}$ whose network can be either similar or different from $f$'s architecture.

Table 4: Clean and attack performance on other models. Red values represent the relative performance differences in attacks under the Marksman's threat model (i.e., full control of the training process).

| Dataset | $f$ : PreActResnet18 | | | | $f$ : Vgg11 | | | |
| --- | --- | --- | --- | --- | --- | --- | --- | --- |
| | $\hat{f}$ : PreActResnet18 | | $\hat{f}$ : Vgg11 | | $\hat{f}$ : Vgg11 | | $\hat{f}$ : PreActResnet18 | |
| | Clean | Attack | Clean | Attack | Clean | Attack | Clean | Attack |
| CIFAR10 | 0.935 | 1.000 | 0.904 | 0.999 | 0.911 | 0.991 | 0.939 | 0.996 |
| | *0.005* | *0.010* | *0.004* | *0.008* | *0.007* | *0.005* | *0.001* | *0.006* |
| GTSRB | 0.986 | 0.999 | 0.973 | 0.997 | 0.976 | 0.995 | 0.987 | 0.996 |
| | *0.008* | *0.006* | *0.006* | *0.004* | *0.006* | *0.007* | *0.007* | *0.003* |

Table 4 shows the clean and attack performance when $T$ is jointly trained with a classifier ($f$), e.g., PreActResnet18, and $\hat{f}$ is another classifier with the same network architecture, i.e., PreActResnet18, or with a different network architecture, i.e., Vgg11, either of which is trained from scratch on the poisoned data created by $T$. It can be seen that Marksman is still effective: nearly optimal clean-data performance and almost 100% ASR. This transferability of $T$ after it is learned is very interesting. We conjecture that the training process of $T$ is related to approximating adversarial example generation of $f$ that shows decent transferability [27, 43], while the detailed investigation is beyond the scope of this paper. Note that the trigger is produced by the trigger generator/function that is the secret held by the adversary, which is qualitatively different from adversarial examples.

### 5.3 Performance against Defensive Measures

In this section, we evaluate the backdoor-injected classifiers by the proposed Marksman framework against popular defensive mechanisms, including Neural Cleanse (model mitigation defense) [45], STRIP (detection based defense) [13], and Spectral Signature (latent-space inspection) [44]. It is important to note that these defenses are originally designed for conventional backdoor attacks with a weaker capability that are only able to misclassify an input to a specific target class.

### 5.3.1 Model Mitigation: Neural Cleanse

We evaluate the robustness of Marksman against Neural Cleanse, a widely-used defensive method based on the pattern optimization approach. Specifically, for every possible label, Neural Cleanse learns the optimal trigger pattern to attack this target label. It then quantifies whether any of the learned trigger patterns is an outlier via a metric called Anomaly Index. Neural Cleanse determines if a model has a backdoor if the Anomaly Index is greater than 2; otherwise, the model is benign.

The results of the benign classifier and the backdoor-injected classifier by Marksman are presented in Figure 3. It can be observed that Marksman bypasses the Neural Cleanse on all datasets.

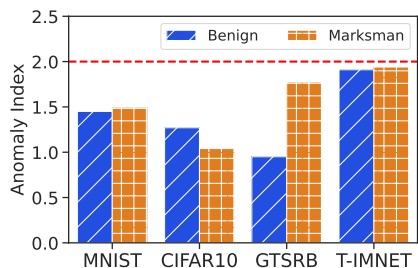

Figure 3: Performance against Neural Cleanse. The benign model is trained without backdoor attacks. A model with Anomaly Index > 2 is considered to be backdoor-injected.

### 5.3.2 Input Perturbation: STRIP

Next, we evaluate the robustness of Marksman against STRIP [13], a representative inference-phase detection method, which perturbs the input image and calculates the entropy of the predictions of these perturbed images. STRIP determines if an image is a backdoor input by relying on the fact that the entropy of the perturbed backdoor input tends to be lower than that of the clean one.

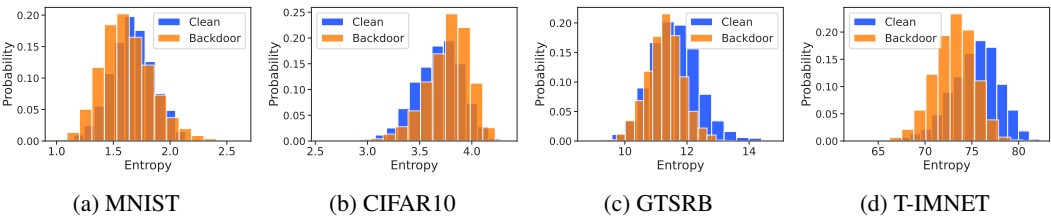

(a) MNIST  (b) CIFAR10  (c) GTSRB  (d) T-IMNET

Figure 4: Performance against STRIP.

We plot the entropy of clean and backdoor images computed by STRIP in Figure 4. As we can observe from this figure, the entropy distributions of the backdoor and clean samples are similar. Thus, Marksman can also bypass the STRIP defense.

### 5.3.3 Latent-space Inspection: Spectral Signature

Spectral Signature [44] is another representative defensive approach that inspects the latent space of the trained classifier. Spectral Signature first finds the top-right singular vector of the covariance matrix of the latent vectors using a small subset of clean samples. Then it calculates the correlation of each sample to this singular vector. Those with the outlier scores are identified as backdoor samples.

We follow the same experiments in [44] and select 5,000 clean samples and 500 backdoor samples for each dataset. Then, we calculate the correlation scores and plot the histograms of the scores for both sets of samples. As we can observe in Figure 5, there is no clear separation between the correlation scores of the backdoor samples and the clean samples, validating the stealthiness of the proposed Marksman backdoor.

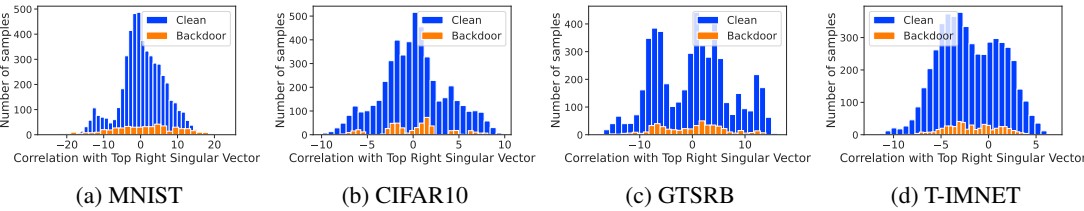

(a) MNIST  (b) CIFAR10  (c) GTSRB  (d) T-IMNET

Figure 5: Performance against Spectral Signature.

### 5.3.4 Performance against Fine-Pruning

We evaluate the robustness of Marksman against Fine-Pruning [26], which is a model analysis based defense. Fine-Pruning finds a learned classifier's dormant neurons (with very low activations) given a small clean dataset. Then it gradually prunes these dormant neurons to mitigate the backdoor without affecting the inference accuracy.

We evaluate Marksman against Fine-Pruning by plotting the clean-data accuracy and ASR when different numbers of the dormant neurons are pruned in Figure 6. We can observe that the ASR drops considerably less than the drop in clean-data performance at all pruning ratios. This suggests that the Fine-Pruning is ineffective against Marksman.

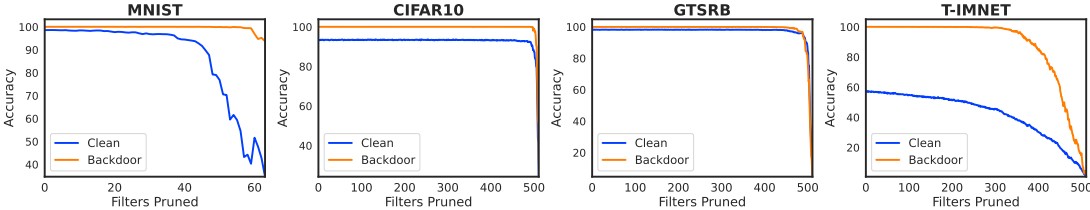

Figure 6: Performance against Fine-Pruning.

## 5.4 Visualization of Backdoor Attack Samples from Marksman

This section presents the backdoor samples with different attack target classes that are different from the true labels (class '2' for MNIST, class '0'-*plane* for CIFAR10), as shown in Figure 7. We can generate a different trigger for each target class to activate the corresponding payload. We can observe that Marksman's attack images are visually indistinguishable from the original images for all target classes, even with such an enhanced adversarial capability.

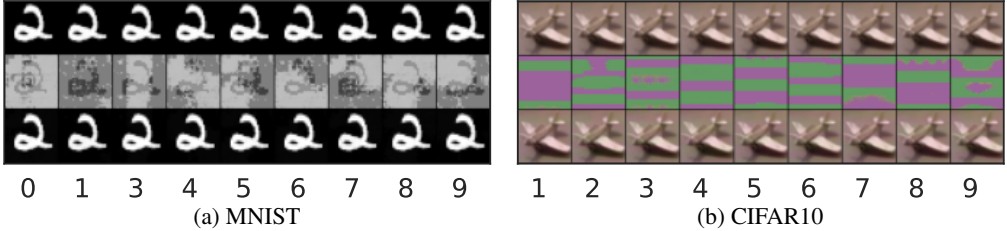

Figure 7: Sample attack images: Clean (Original), Residual (Amplified 50×), and Backdoor samples in 1st, 2nd and, 3rd rows, respectively. The target attack labels are in the last rows.

## 6 Conclusion

In this paper, we proposed a novel backdoor attack, Marksman, with a much more powerful payload where the adversary can arbitrarily choose the target class given any input during inference. To the best of our knowledge, Marksman is the first multi-trigger and multi-payload backdoor attack method that can misclassify an input to any target class. This framework can effectively and efficiently learn the trigger function and inject the backdoor into a DNN model. Given a target label, the proposed class-condition generative trigger function can generate an imperceptible trigger pattern to cause the model to predict the target label. We then propose a constrained optimization objective that can simultaneously learn the trigger function and poison the model. Our experimental results over popular datasets demonstrate the effectiveness of Marksman and the stealthiness against the existing representative defenses. Our work takes another significant step toward understanding the extensive risks of backdoor attacks in practice. This work calls for defensive studies to counter Marksman's more powerful yet sophisticated multi-trigger and multi-payload attacks.

## Acknowledgement

We thank anonymous Referees for their constructive comments. The research was conducted while all authors worked at the Baidu Cognitive Computing Lab, 10900 NE 8th St. Bellevue, WA 98004, USA.

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
