This document provides additional details, analysis, and experimental results. We begin by discussing the detailed experimental setup and implementation of the methods in Section 7. Then, we provide additional empirical experiments in Section 8. Finally, we discuss the limitation of our work in Section 9.

# 7 Detailed Experimental Setup

To evaluate our method, we use four datasets, MNIST, CIFAR10, GTSRB (German Traffic Sign Recognition Benchmark), and T-IMNET, to evaluate our method. Note that MNIST, CIFAR10, and GTSRB have been widely used in the literature of backdoor attacks on DNN. On the other hand, the use of a more complex dataset with more classes, T-IMNET, enables better evaluation for the scalability of multiple-trigger and multi-payload backdoor. We present the details of the our experiments on these datasets below:

- **MNIST** [21]: We applied random cropping and random rotation as data augmentation for the training process. During the evaluation stage, no augmentation is applied. The dataset can be found in: http://yann.lecun.com/exdb/mnist

- **CIFAR10** [18]: We applied random cropping, random rotation, and random horizontal flilp as data augmentation for the training process. The dataset can be found in: https://www.cs.toronto.edu/ ~kriz/cifar.html

- **GTSRB** [40]: In our experiments, GTSRB input images are all resized into $32 \times 32$ pixels, then applied a similar data augmentation as that of CIFAR10 in training. In the evaluation stage, no augmentation is used. The dataset can be found in: http://benchmark.ini.rub.de/?section=gtsrb&subsection=dataset

- **T-IMNET** [47]. Input images are all resized into $64 \times 64$ resolution. A similar data augmentation as that of CIFAR10 is applied in the training stage. No augmentation is used in the evaluation stage. The dataset can be found in: http://cs231n.stanford.edu/tiny-imagenet-200.zip

Our experiments are conducted in multiple Linux machines, each of which has 128 cores, 1536GiB of RAM, 8 A100-PCIE-40GB GPUs. The CUDA version is 11.2.

## 7.1 Class-Conditional Trigger Generator

For all experiments of Marksman, we model the class-conditional noise generator (trigger-pattern generator) $g$ as an autoencoder, whose input is an image and a learnable embedding of a target class. The architecture of the autoencoder is presented in Table 5.

Table 5: Autoencoder-based class-conditional trigger-pattern generator network. The size of the class embedding vector equals the number of possible labels (i.e., 10 for MNIST and CIFAR10, 43 for GTSRB, and 200 for T-IMNET).

| Layer | Filters | Filter Size | Stride | Padding | Activation |
|---|---|---|---|---|---|
| Conv2D | 16 | $3 \times 3$ | 3 | 1 | BatchNorm2D+ReLU |
| MaxPool2d | - | $2 \times 2$ | 2 | 0 | - |
| Conv2D | 64 | $3 \times 3$ | 2 | 1 | BatchNorm2D+ReLU |
| MaxPool2d | - | $2 \times 2$ | 2 | 0 | - |
| ConvTranspose2D | 128 | $3 \times 3$ | 2 | - | BatchNorm2D+ReLU |
| ConvTranspose2D | 64 | $5 \times 5$ | 3 | 1 | BatchNorm2D+ReLU |
| ConvTranspose2D | 1 | $2 \times 2$ | 2 | 1 | BatchNorm2D+Tanh |

## 7.2 Classification Models

For MNIST, we use the same simple CNN classifier as in WaNet [35] (detailed in Table 6). As mentioned in the main paper, we use PreActResnet18 [16] for CIFAR10 and GTSRB datasets, and Resnet18 [16] for T-IMNET.

Table 6: CNN architecture for MNIST.

| Layer | Filters | Filter Size | Stride | Padding | Activation |
|-------|---------|-------------|--------|---------|------------|
| Conv2D | 32 | $3 \times 3$ | 2 | 1 | ReLU |
| Conv2D | 64 | $3 \times 3$ | 2 | 0 | ReLU |
| Conv2D | 64 | $3 \times 3$ | 2 | 0 | ReLU |
| Linear | 512 | - | - | - | ReLU |
| Conv2D | 10 | - | - | - | Softmax |

## 7.3  Training Hyperparameters

Table 7 provides additional details to Section 5.1 in the main paper.

Table 7: Experimental setup and parameters for the datasets we used in this paper.

| | MNIST | CIFAR10 | GTSRB | T-IMNET |
|---|-------|---------|-------|---------|
| Optimizer | SGD | SGD | SGD | SGD |
| Batch Size | 128 | 128 | 128 | 128 |
| Learning Rate | 0.01 | 0.01 | 0.01 | 0.01 |
| Learning Rate Schedule | 10,20,30,40 | 100,200,300,400 | 100,200,300,400 | 100,200,300,400 |
| Learning Rate Decay | 0.1 | 0.1 | 0.1 | 0.1 |
| Training Epochs | 50 epochs | 500 epochs | 500 epochs | 500 epochs |
| Clean Accuracy | 0.99 | 0.94 | 0.99 | 0.58 |

As indicated in Section 5.1 of the main paper, we perform grid searches to select the best $\alpha$ and $\beta$ values using the CIFAR10 dataset. The selected values are $0.8$ for $\alpha$ and $1.0$ for $\beta$. We use these values for all the experiments

# 8   Additional Experimental Results

We first provide, in Section 8.1, additional attack performance when varying the poisoning rate for ReFoolMT and WaNetMT, as well as for PatchMT on T-IMNET. In Section 8.2, we provide a qualitative analysis of the hyperparameters $\alpha$ and $\beta$ on the CIFAR10 dataset. Finally, we present the statistical errors for the main experiments of the paper in Section 8.3.

## 8.1   Additional Clean & Attack Performance Results with Different Poisoning Rates

We present the attack performance while varying the poisoning rates for PatchMT on T-IMNET in Figure 8, and the other baselines (ReFoolMT and WaNetMT) in Figure 9. We can observe a similar phenomenon (i.e., either performs noticeably worse on clean data with higher attack performance or preserves the clean-data performance with significantly lower ASR) for PatchMT on T-IMNET in Figure 8 and for ReFoolMT on all the datasets in Figure 9. For WaNetMT, its clean-data performance is consistently sub-optimal, suggesting that it is more difficult to perform the multi-trigger and multi-payload attacks using the warping mechanism.

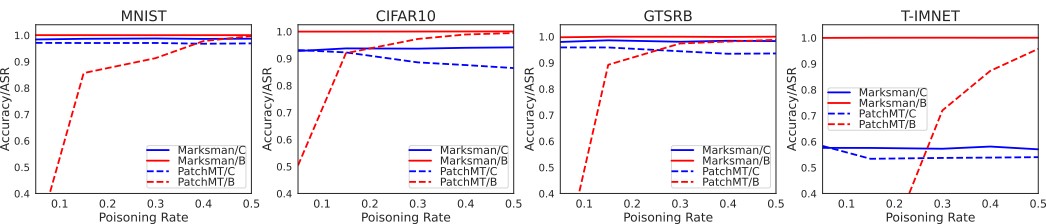

Figure 8: Clean (**/C**) and attack (**/B**) performance with different poisoning rates for Marksman and PatchMT.

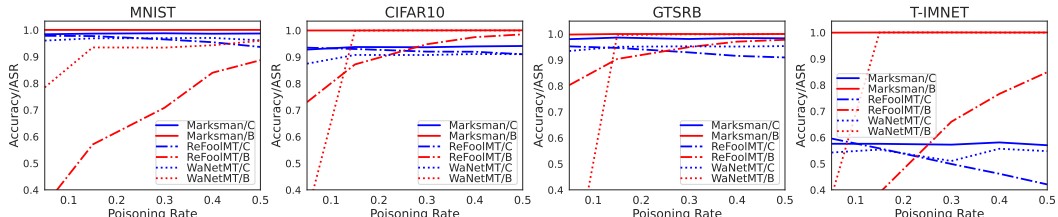

Figure 9: Clean (**/C**) and attack (**/B**) performance with different poisoning rates for Marksman, ReFoolMT, and WaNetMT.

## 8.2 Hyperparameter Analysis

As indicated in Section 5.1 of the main paper, we perform grid searches to select the best $\alpha$ and $\beta$ using the CIFAR10 dataset and use these values for all the experiments. However, we additionally present the qualitative analysis of the hyperparameters $\alpha$ and $\beta$ in Figure 10. As we can observe, lower values of $\alpha$ result in lower clean-data accuracy, while an optimal value is closer to 1.0 (around 0.8). Note that when $\alpha = 1.0$, the classifier is not poisoned. Thus, the ASR drops to 0. On the other hand, clean and attack performances are not very sensitive to the values of $\beta$. In our experiments, we set $\beta = 1.0$.

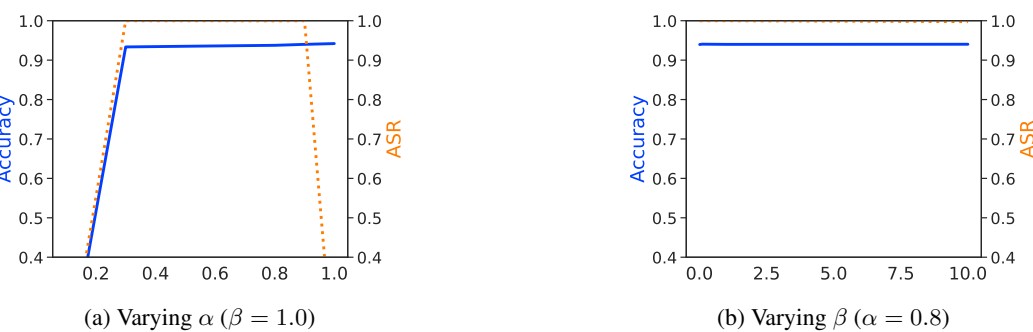

(a) Varying $\alpha$ ($\beta = 1.0$)

(b) Varying $\beta$ ($\alpha = 0.8$)

Figure 10: Clean and attack performance when varying $\alpha$ and $\beta$ (while keeping the other fixed) on CIFAR10 dataset.

## 8.3 Statistical Importance

We present the statistical errors in our experiments in Figure 11. As we can observe, most methods, except for ReFoolMT on MNIST, have very narrow (99%) confidence intervals in all experiments. We also verify that the superior performance of Marksman is statistically significant with a p-value less than 0.01.

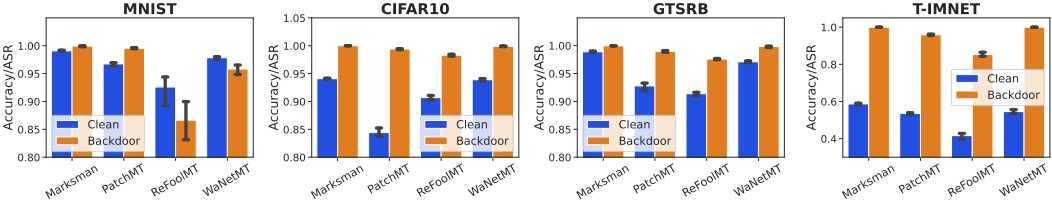

Figure 11: Performance and the error bars (confidence intervals).

## 8.4 Performance against NAD and ANP

We evaluate the robustness of Marksman against two representative post-training defenses, Neural Attention Distillation (NAD) [23] and Adversarial Neuron Pruning (ANP) [46]. We follow the same

experimental settings of NAD and ANP, which assumes a small subset of clean data (5%). Table 8 show the clean-data Accuracy and ASR of the poisoned models after they undergo the backdoor erasing processes of NAD and ANP. As we can observe, for NAD, while ASR decreases, ACC performance degrades even more significantly. This demonstrates that NAD's defense is ineffective against Marksman's attack. For ANP, the defensive process does not significantly degrade the ASRs of Marksman, while the ACCs also drop more significantly. Again, ANP is not effective against Marksman's attack.

Table 8: Performance against NAD and ANP.

| Dataset | NAD | | ANP | |
|---|---|---|---|---|
| | Clean | Attack | Clean | Attack |
| CIFAR10 | 0.365 | 0.069 | 0.938 | 0.647 |
| GTSRB | 0.114 | 0.022 | 0.890 | 0.651 |

# 9 Limitations

To the best of our knowledge, Marksman is the first work that studies multi-trigger and multi-payload backdoor with the capability of misclassifying an input to any target class. However, this assumes that the adversary has the ability to embed digitally generated triggers into the images before feeding them to the classifier. An interesting future direction is to extend the multi-trigger and multi-payload scenario into physical attacks. Such evaluations can further assess whether Marksman is only a hypothetical phenomenon or indeed a real-world threat.

Since this is the first work in this direction, there is no existing defense that targets the scenario of Marksman, which can also be observed from our experiments that the existing representative defensive methods are not effective against Marksman. On the other hand, these experiments do not reasonably or effectively evaluate the performance of our approach. To this end, we encourage future research on developing more powerful defenses to combat our stealthy backdoor attack with significantly enhanced adversarial capabilities.

**Societal Impacts:** Our work on the backdoor attack is likely to increase the awareness and understanding of such vulnerability on neural networks. The proposed attack, if not appropriately used, may bring security threats to the existing DNN applications. We believe our study is an important step towards understanding the full capability of backdoor attacks. This knowledge will, in turn, facilitate the further development of secure and trustworthy DNN models and powerful defensive solutions.