# OpenReview forum: "Marksman Backdoor: Backdoor Attacks with Arbitrary Target Class"
_NeurIPS.cc/2022/Conference — NeurIPS 2022 Accept_

### Official Review · Reviewer_a9D8 · 2022-07-10

**Rating:** 5
**Confidence:** 5
**Soundness:** 3 good
**Presentation:** 2 fair
**Contribution:** 2 fair

**Summary:**

In this work, the authors investigate a novel backdoor attack paradigm, Marksman, where the adversary can arbitrarily choose which target class the model will misclassify given any input during inference. They use a class-conditional generative model to represent the trigger function and inject the backdoor in a constrained optimization framework. They verify their attack can achieve a high attack success rate in the experiments.

**Questions:**

As listed in Weakness

**Ethics Review Area:**

["I don’t know"]

**Limitations:**

As listed in Weakness. I think that this is an interesting paper and score can be improved if concerns listed above are resolved.

**Strengths And Weaknesses:**

## Strength

- The paper is well written and easy to follow.
- The conclusions are backed up with proper experiments.
- The authors proposed a novel backdoor attack paradigm, Marksman. This is the first multi-trigger and multi-payload backdoor attack with the capability of misclassifying an input to any target class.
- The authors experimentally proved that Marksman can be a threat in the ML community.

## Weakness

While this is an interesting paper, I find that there are several weaknesses:

1.  Although Marksman is the first work that studies multi-trigger and multi-payload backdoor with the capability of misclassifying an input to any target class, it is not clear what is the motivation behind Marksman for me. The authors state that Marksman can arbitrarily choose which target class the model will misclassify given any input during inference, more powerful than existing attacks. However, what is the advantage of Marksman? Why do we need Marksman? Since this is the main part of the paper, I would expect that there’s a more convincing discussion about the necessity of the proposed Marksman.
2. Only some early defenses are considered. However, there are more advanced backdoor defenses recently, e.g. NAD [1], ANP [2].

[1] Yige Li, et al. "Neural Attention Distillation: Erasing Backdoor Triggers from Deep Neural Networks", ICLR 2021.

[2] Dongxian Wu, et al. "Adversarial Neuron Pruning Purifies Backdoored Deep Models", NeurIPS 2021.

-------------------------------------------------------------------------------------------------------------------------------------------------

I appreciate the authors for the detailed response, which addresses most of my concerns. However, the formulation and optimization of Marksman is inspired by LIRA [1] and thus the technical novelty is limited. Besides, in my opinion, the multi-target setting in this work seems not very practical. Putting it all together, I decided to increase my score to 5.

[1] Khoa Doan, et al. "LIRA: Learnable, Imperceptible and Robust Backdoor Attacks", ICCV 2021

---

> ### Author Response · Authors · 2022-08-01
> **Thank you the reviewer for the thorough reading of our paper and valuable suggestions.**
>
> **Q1: Although Marksman is the first work that studies multi-trigger and multi-payload backdoor with the capability of misclassifying an input to any target class, it is not clear what is the motivation behind Marksman for me. The authors state that Marksman can arbitrarily choose which target class the model will misclassify given any input during inference, more powerful than existing attacks. However, what is the advantage of Marksman? Why do we need Marksman? Since this is the main part of the paper, I would expect that there’s a more convincing discussion about the necessity of the proposed Marksman.**
>
> Marksman extends the adversarial capability of the backdoor attack. In the setting of previous works, for a given input, the adversary can only inject the trigger to force the classification to a particular target label. In contrast, using Marksman, the adversary will be able to generate the trigger for any target label, which makes it more potent. Similar motivations are also shared by prior works [40,41], although the number of possible target backdoor classes is still limited, i.e.,3 or 4. Following the popular example in backdoor attack studies, Marksman has the capability to change the classification from “stop sign” to “speed limit”, “bicycle crossing”, “give way”, “no u-turn”, “no vehicles”, “turn left”, or “mandatory roundabout”, under different situations/times/conditions, while the setting of previous works may only change from “stop sign” to “speed limit”. We will expand this discussion in the later version.
>
> **Q2: Only some early defenses are considered. However, there are more advanced backdoor defenses recently, e.g. NAD [1], ANP [2].**
>
> Thanks for bringing these recent works to our attention. As suggested, we perform additional experiments against these two defenses. The results are shown below. We follow the same experimental settings of NAD and ANP, which also assumes a small subset of clean data (5%). The poisoned models crafted by Marksman undergo the blackdoor-erasing process of NAD and ANP and are then evaluated on both clean and backdoor samples. Similarly, we reported the ASR and Accuracy (ACC) of the processed models. As we can observe, for NAD, while ASR decreases, ACC performance degrades even more significantly. This demonstrates that NAD’s defense is ineffective against Marksman’s attack. For ANP, the defensive process does not significantly degrade the ASRs of Marksman, while the ACCs also drop more significantly. Again, ANP is not effective against Marksman’s attack. Similar to our discussion in the main paper for other defenses, we conjecture that the ability to bypass several existing backdoor defenses is due to the fact that Marksman’s attack poison the model in a very different way than the previously studied backdoor attacks. This calls for more defensive studies to counter the more powerful yet sophisticated multi-trigger and multi-payload attacks of Marksman.
>
> Performance Against NAD Defense
>
> | Dataset | ASR    | ACC    |
> |-------- | ------ | -------|
> | CIFAR10 | 0.3654 | 0.0695 |
> | GTSRB   | 0.1144 | 0.0220 |
>
> Performance Against ANP Defense
>
> | Dataset | ASR    | ACC    |
> |-------- | ------ | -------|
> | CIFAR10 | 0.9385 | 0.6474 |
> | GTSRB   | 0.8901 | 0.6512 |

---

> ### Author Response · Authors · 2022-08-06
> **Thank you for your valuable comment post rebuttal**
>
> Thank you for reading our response and improving the paper’s rating. We’d like to clarify the novelty of our work, which we think are significant contributions to the backdoor domain.
>
> While the computational technique of Marksman has some similarities to LIRA, and in fact, most input-aware attacks, our work focuses on studying a completely new and extremely sophisticated backdoor attack, i.e., multi-trigger multi-payload attack, that has not yet been previously studied in the backdoor literature. As also discussed in Section 1, the primary contribution of our work is this novel type of backdoor attacks. Different from the existing attacks that can only target one single label for a given input, Marksman’s attack can arbitrarily choose any label to target during inference, which makes Marksman significantly more harmful than the existing attacks. Nevertheless, our work relies on jointly poisoning the classifier and optimizing a class-conditional generative trigger function, which is different from the existing input-aware attacks where the trigger function is optimized independently from the classifier or is unconditional. Furthermore, Marksman’s alternating learning algorithm only involves one stage, which is different from LIRA’s learning algorithm which includes two stages. Finally, we empirically demonstrated the danger of Markman’s multi-trigger multi-payload attack: almost perfect ASRs and pristine clean-data accuracy on multiple representative datasets. More importantly, we showed that Marksman is able to easily bypass several representative defensive mechanisms from different categories. We conjectured that Marksman’s novel attack with real-world harmful consequences might poison the model in a very different way than these existing backdoor attacks, which calls for further studies to counter this type of sophisticated threat.
>
> In summary, the study of a completely new type of backdoor attacks, the proposed learning model and algorithm, and the extensive analysis and insight of the proposed attacks are the contributions of our work. This is different than LIRA and more broadly all existing input-aware attacks that focus on imperceptibility of the single-target single payload attacks.

---

### Official Review · Reviewer_jzLs · 2022-07-10

**Rating:** 4
**Confidence:** 4
**Soundness:** 3 good
**Presentation:** 3 good
**Contribution:** 2 fair

**Summary:**

This paper proposes a backdoor attack called Marksman, with a more powerful
payload against deep neural networks. Specifically, it proposes to optimize a
trigger generative function to find an optimal trigger pattern for an
arbitrary target class while simultaneously injecting a generative backdoor
into the model. The results show that Marksman can attack any class at the
adversary's will during inference time, and it can bypass three existing
defenses.


**Questions:**

What is the fundamental difference between this work and existing works? This
is not well explained in the paper. Nguyen et al. "Input-aware dynamic
backdoor attack" from NeruIPS 2020 also consider a generic backdoor that is
input-specific. The paper seems to be a simple extension of this work in
multiple target class settings.

The technique is an improved version of GAN-based trigger generation
methods [8,9,28]. But during evaluation, it chooses to compare
patch-based trigger patterns from [13] as its baseline. Is it possible to
extend the current GAN-based methods [8,9,23] and inject them into the model?
It seems to be a better baseline.

Moreover, the argument that injecting multiple triggers into a model will lead
to more significant perturbations has never been justified by experiments. Existing
works try to combine adversarial examples with backdoors to create invisible
triggers. Following this line of work, is it possible that multiple triggers
can have a reasonable size similar to a single one? GAN-based
method potentially can do the same (depending on the optimization results).
But it needs to be justified by the paper as it is making a claim here.

The work is only evaluated on post-training backdoor investigation methods.
There are other types of defenses, e.g., training time defense (NAD, MOTH)
and data cleaning as a defense (anti-backdoor learning). The paper does not evaluate
these defenses. Adaptive defense is also not considered.

**Limitations:**

The limitation discussion focuses on backdoor "scanning" techniques, but there are more types of defenses under this threat model. I encourage the authors to discuss them with experiments.

**Strengths And Weaknesses:**

Strengths:

Backdoor is a trendy topic. The paper is well written and states its
constrained optimization technique clearly.

Weaknesses:

The difference between its proposed attack method and existing works is not
clearly explained. The comparison with baseline methods is limited. It lacks
ablation studies on other important parameters like learning rate and training
epoch.

---

> ### Author Response · Authors · 2022-08-01
> **We thank the reviewer for the insightful and valuable comments.**
>
> **Q1: What is the fundamental difference between this work and existing works?**
>
> Similar to WaNet and LIRA, our work is an input-aware attack. However, we showed that simply extending the existing works, including input-aware attack (as shown in WaNetMT), will not result in good ASR while simultaneously preserving the clean-data accuracy. Please see our results in Section B.2 of the supplementary material (Tables 8, 9, and 10). These extensions will lead to a much larger model perturbation than the single-trigger and single-payload attack. As the number of target classes increases, both the ASR and the clean accuracy will be significantly degraded. In contrast, Marksman jointly optimizes the multi-trigger and multi-payload trigger function (via a class-conditional generative model) and the poisoned classifier in order to achieve both higher ASR and clean accuracy.  As also suggested by other reviewers, we will reorganize the paper and move the comparisons with ReFoolMT and WaNetMT to the main paper.
>
> **Q2: Is it possible to extend the GAN-based methods [8,9,23] and inject them into the model?**
>
> Thank you for the observation. Please see our results in the supplementary material for more baselines (ReFoolMT and WaNetMT). Our reason for not considering GAN-based triggers is due to efficiency. For example, if we follow a similar approach as in PatchMT, RefoolMT, and WaNetMT to extend the work in [8], we are required to train multiple trigger generators, one for each target class. Due to the complexity of GAN training, it is not practical as the number of classes increases (10 for CIFAR10 and MNIST, 43 for GTSRB, and 200 for TinyImageNet). Instead, Marksman tackles this challenge by modeling the trigger function as a class-conditional generative model. Furthermore, we also showed that jointly optimizing the trigger and the classifier is crucial in carrying out the proposed attack, but some GAN-based methods [8,23] do not simultaneously train the classifier.
>
> **Q3: The argument that injecting multiple triggers into a model will lead to more significant perturbations has never been justified by experiments. Existing works try to combine adversarial examples with backdoors to create invisible triggers: is it possible that multiple triggers can have a reasonable size similar to a single one?**
>
> Thanks for the comments and insightful ideas. Injecting multiple backdoors, each for one of the target classes would inevitably tweak the model more significantly than only injecting the backdoor for one class. In our humble opinion, multiple triggers should have more perturbations than a single one, if the single one is carefully optimized. Our results in Tables 1, 2, 3, 8, 9, and 10 also proved this. For example, injecting a very small percentage of poisoned samples for each trigger results in very low ASRs, while injecting a larger number of poisoned samples to improve the ASRs inevitably leads to a non-trivial drop in clean performance. To the best of our knowledge, Marksman is the first backdoor method with the capability of misclassifying an input to any target class. Marksman simultaneously learns a class-conditional generative model as the trigger function and poisons the classifier to ensure that both optimal ASRs (almost 100% in all experiments) and optimal clean-data accuracies (that are almost the same as those of the benign models) can be achieved.  We consider the idea of leveraging adversarial examples has the potential to further optimize multiple triggers, which could inspire interesting future works.
>
> **Q4: The work is only evaluated on post-training backdoor investigation methods. There are other types of defenses, e.g., training time defense (NAD, MOTH) and data cleaning as a defense (anti-backdoor learning).**
>
> As we indicated in Section 3, we consider the threat model where the adversary has full control over the training process, which is consistent with prior works of input-aware attacks [8,28,9]. Thus, the training time defense and data cleaning (i.e., anti-backdoor learning), which assume that the victim controls the training process, do not align with our threat model. Nevertheless, we provide more experiments on NAD (which is actually a post-training backdoor erasing defense) and ANP, as also suggested by Reviewer a9D8. Both NAD and ANP defenses are not effective against Marksman. We conjecture that the ability to bypass several existing backdoor defenses is due to the fact that Marksman’s attack poisons the model in a very different way than the previously studied backdoor attacks. This calls for more defensive studies to counter the more powerful yet sophisticated Marksman attack.
>
> Performance Against NAD Defense
>
> | Dataset | ASR    | ACC    |
> |-------- | ------ | -------|
> | CIFAR10 | 0.3654 | 0.0695 |
> | GTSRB   | 0.1144 | 0.0220 |
>
> Performance Against ANP Defense
>
> | Dataset | ASR    | ACC    |
> |-------- | ------ | -------|
> | CIFAR10 | 0.9385 | 0.6474 |
> | GTSRB   | 0.8901 | 0.6512 |

---

> > ### Author Response · Authors · 2022-08-07
> > **Looking forward to your responses or further comments!**
> >
> > Thank you again for your valuable comments. If you have any additional questions about our response, please kindly let us know, and we will try to answer quickly during the discussion period.

---

> > > ### Comment · Reviewer_jzLs · 2022-08-09
> > > **Response**
> > >
> > > Thank you very much for your efforts in answering the questions and addressing the concerns. They will be taken into full consideration.

---

### Official Review · Reviewer_ad2z · 2022-07-11

**Rating:** 5
**Confidence:** 3
**Soundness:** 3 good
**Presentation:** 3 good
**Contribution:** 2 fair

**Summary:**

This paper presents a novel setting for backdoor attacks by injecting different triggers for different labels. It designs a class-condition generative trigger function. Then if given the target label, the well-trained generator T can generate an imperceptible trigger pattern to cause the model to predict the target label.

**Questions:**

1. Experimental results of PatchMT (in Table 3) need to be further explained.
2. Could the authors add some experiments by simply injecting the generated triggers into training data, and then train models from-scratch? The results and comparisons may not be fair enough now.

**Limitations:**

Yes,  the authors have adequately addressed the limitations and potential negative societal impact of their work.

**Strengths And Weaknesses:**

+ The paper is generally well written and easy to understand.
+ The proposed method is interesting and has high novelty.
+ Datasets are adequate for backdoor attacks.

- The experimental results of PatchMT in Table 3 seem quite confusing for me. As far as I understand, injecting 10 different trigger patterns referring to 10 different labels in CIFAR-10 should have still high ASRs even with 10% poisoned images, in which situation 1% data is poisoned referring to one class. BadNets has shown injecting 1% poisoned data should have a high ASR.
- I have doubts about the poisoning procedure. I do understand the proposed Marksman needs to optimize T and then generate triggers using T. But most backdoor baseline methods need to only inject poisoned data and do not control the training procedure. So I would like to see the results using well-trained T to generate triggers and inject them into clean data, then train the backdoored models either from scratch or fine-tuning.

---

> ### Author Response · Authors · 2022-08-01
> **We thank the reviewer for the insightful and valuable comments.**
>
> **Q1: Experimental results of PatchMT (in Table 3) need to be further explained. The experimental results of PatchMT in Table 3 seem quite confusing for me. As far as I understand, injecting 10 different trigger patterns referring to 10 different labels in CIFAR-10 should have still high ASRs even with 10% poisoned images, in which situation 1% data is poisoned referring to one class. BadNets has shown injecting 1% poisoned data should have a high ASR.**
>
> As we discussed in Section 5.1, PatchMT is a multi-trigger, multi-payload extension that is based on BadNets. PatchMT uses patch-based triggers (with similar designs as in BadNets) with different patterns for different target classes. BadNets can yield a high ASR by injecting 1% poisoned data. However, injecting such 1% poisoned data for each of the classes would inevitably tweak the model more significantly than only injecting 1% poisoned data for one class. Meanwhile, injecting more poisoned data with different target labels would also reduce the attack performance for one particular target label, under the setting of PatchMT. Furthermore, this strategy (i.e., injecting 1% poisoned data for one class) does not work in other datasets with a higher number of classes, such as GTSRB (43 classes) and TinyImagenet (200 classes).
>
> We also consider different trigger designs, as in RefoolMT and WaNetMT. We show that these extensions cannot carry out effective multi-trigger, multi-payload attacks as can be seen in Marksman.
>
> **Q2: Could the authors add some experiments by simply injecting the generated triggers into training data, and then train models from-scratch? The results and comparisons may not be fair enough now. I have doubts about the poisoning procedure. I do understand the proposed Marksman needs to optimize T and then generate triggers using T. But most backdoor baseline methods need to only inject poisoned data and do not control the training procedure. So I would like to see the results using well-trained T to generate triggers and inject them into clean data, then train the backdoored models either from scratch or fine-tuning.**
>
> As indicated in Section 3, we consider the threat model where the adversary has full control over the training process, which is also considered in various related works of input-aware attacks [8,28,9]. This threat model is also common in practice, where the end-users outsource their model-building effort to the MLaaS companies due to the complexity of training such models. Nevertheless, we also provided an experiment in Section 5.2.3 where the adversary can only inject poisoned samples generated by the learned trigger function into the training data and the model training process is performed by the victim. As can be seen in this experiment, our attack is still quite successful.

---

> > ### Comment · Reviewer_ad2z · 2022-08-06
> > **Thank you for the kind response.**
> >
> > The authors addressed most of my concerns. I also read the authors' responses for other reviewers and went through the manuscript again. Overall, it is an interesting paper. And the proposed conditional multi-label multi-trigger setting is quite novel. After that, I would like to raise my score.
> >
> > Just for discussion, I am curious about the following two points:
> > 1. How will the architecture of $g$ affect in this attack? Does $g$ need to be quite complex as it needs to modify the triggers for large-scale dataset, e.g., TinyImageNet?
> > 2. What about the practical overhead for Marksman? As the author mentioned, it can be deployed in real world scenarios like MLaaS, I would like to further compare its time or computational cost with standard model training without any poisoning.

---

> > > ### Author Response · Authors · 2022-08-07
> > > **Our response to the additional questions**
> > >
> > > **Q1: How will the architecture of g affect in this attack? Does g need to be quite complex as it needs to modify the triggers for large-scale dataset, e.g., TinyImageNet?**
> > >
> > > Thank you for your great question. In prior related works, some encoder-decoder architectures, such as UNet for ImageNet and a similar autoencoder as in the model described in the supplementary material for CIFAR10/GTSRB, are used. In our experiments, we observe the same autoencoder can be used for all the evaluated datasets, and Marksman's attacks can already reach optimal performance.
> > >
> > > **Q2: What about the practical overhead for Marksman? As the author mentioned, it can be deployed in real world scenarios like MLaaS, I would like to further compare its time or computational cost with standard model training without any poisoning.**
> > >
> > > We sincerely appreciate this insightful question. As described in Section 4.4 and Algorithm 1, at each iteration, in addition to the standard training of the classifier, we need to update the trajectory of the trigger function. Consequently, the computational cost of model training with poisoning will be more than that of standard training. For example, with Tesla P100 and Intel(R) Xeon(R) CPU E5-2630 v4 @ 2.20GHz, on CIFAR10, we observe that training with poisoning takes an average of 89 seconds per epoch, while it is 40 seconds per epoch in standard training.
> > >
> > > While the computational cost of Marksman's optimization is higher than that of standard training, It is possible to train T and f jointly for several epochs, then fix T and generate the poison training data and finally continue to train only f on such poison training data. As described in Section 5.2.3, this approach also works well while the ASRs only drop slightly. However, the training time of Marksman will be significantly decreased.

---

### Official Review · Reviewer_Kg8P · 2022-07-14

**Rating:** 6
**Confidence:** 4
**Soundness:** 3 good
**Presentation:** 4 excellent
**Contribution:** 3 good

**Summary:**

The paper introduces a novel backdoor attack against machine learning classifiers by systematizing the generation of the triggers through a class-conditional generative model which allows to introduce backdoors targeting an arbitrarily large number of classes. The attack assumes that the attacker is in control of the training process, so that the model is trained by optimizing a loss function that includes the learning of the generative model for the triggers. However, the attack can also be used via data poisoning, by training first a surrogate model to learn the generative model for the backdoor triggers, and then, poisoning the target classifiers using malicious data points generated with this generative model. Different experiments on computer vision benchmarks show the effectiveness of the proposed approach to introduce backdoors with a very high attack success rate and a very small loss in performance.

**Questions:**

+ What are the types of trigger patterns used for performing the PatchMT attack? The type of triggers used for this class can have a significant impact in the effectiveness of the attack, especially, when using data augmentation techniques such as cropping, rotations and flips, and poisoning with low fraction of poisoning examples.
+ In Table 4, what is the drop in performance for the different models evaluated on the clean data?

(I included some suggestions below in the discussions of the limitations of the paper).


**Limitations:**

The approach proposed to insert the backdoors is interesting, as by solving the optimization problem in (3) it seems that the information of the backdoors is hidden better in the parameters of the model resulting after training. This enable two different attack scenarios: one where the attacker is in control of the whole training process (i.e., the attacker optimizes (3) to produce the model); and another scenario where the attacker first optimizes (3) to generate poisons that, then, are injected in the training set so that the training of the model is external to the attacker.
Given this, the comparison in Tables 1 and 2 with PatchMT is somewhat not really fair, as PatchMT just rely on poisoning the training set. For this, I think it would be more appropriate to compare PatchMT in the case where Marksman is used to poison the training dataset (as in Section 5.2.3).
Following the previous point, it is a bit strange that the authors did not report the loss in performance evaluated on the clean datasets in the models evaluated in Table 4. As per in my previous question, it would be useful to know if the loss in performance is higher when the backdoors are inserted via poisoning attacks.
By looking at the results in the appendix, I think that bringing the comparisons with ReFoolMT and WaNetMT to the main paper (there is space in Tables 1 and 2, for example, to do that) would be beneficial to provide a more comprehensive evaluation in Section 5.
About the defenses, the authors provide an interesting evaluation that evidences the limitations of current defenses to defend against Marksman. However, these defenses are designed for backdoors that are typically generated with poisoning attacks. It would be interesting to see if the defenses work to reduce the effectiveness of the Marksman attack when the attack is performed by poisoning the training dataset, and then, training the model. It would be interesting to see if there are significant different between the two threat models and, perhaps, the results can foster the investigation of more sophisticated defensive methods for backdoor attacks where the information of the backdoors is blended better in the parameters of the model.


**Strengths And Weaknesses:**

Strengths:
+ The attack strategy to systematize the backdoor attack allow to target a broader range of classes compared to existing backdoor attacks in the research literature, resulting in a stronger attack.
+ The systematization of the attack by learning the trigger generator and the model at the same time is very interesting and enable the insertion of more stealthy backdoors assuming a threat model where the attacker is in control of the learning process.
+ The transferability of the backdoors, so that malicious samples crafted with the generator can be used to insert backdoors in a different target model seems interesting, although the experiments are not very thorough compared to the rest (e.g. less datasets).
+ The paper is easy to follow and, overall, well written and clear.

Weaknesses:
- Part of the experimental evaluation can be improved, especially as some direct comparisons are not really fair, as the attacker’s capabilities are different (see comments below).
- It is unclear what is the loss in performance when Marksman is used on other models (Section 5.2.3). The evaluation of this aspect is not very thorough.
- The positioning of the paper, in terms of the threat model considered, with respect to other papers in the related work can be improved.

---

> ### Author Response · Authors · 2022-08-01
> **We sincerely thank the reviewer for the thoughtful comments.**
>
> **Q1: What are the types of trigger patterns used for performing the PatchMT attack? The type of triggers used for this class can have a significant impact in the effectiveness of the attack, especially, when using data augmentation techniques such as cropping, rotations and flips, and poisoning with low fraction of poisoning examples.**
>
> As indicated in Section 5.1, PatchMT is a multi-trigger, multi-payload extension that is based on BadNets. PatchMT uses patch-based triggers (with similar designs as in BadNets) with different patterns for different target classes. We keep the same type of trigger as BadNets to generate the baseline comparison to our proposed Marksman. We also consider different representative trigger designs, such as blending-based triggers in RefoolMT and input-aware warping-based triggers in WaNetMT. We show that these extensions cannot carry out effective multi-trigger, multi-payload attacks as can be seen in Marksman. When the poisoning rate is low (e.g., 10%), the ASR is also very low (below 50%). When the poisoning rate is high (e.g., 50%), the ASRs are higher, but the clean-data accuracies drop below acceptable levels.
>
> **Q2: In Table 4, what is the drop in performance for the different models evaluated on the clean data?**
>
> Section 5.2.3 evaluates the transferability of the learned class-conditional trigger function. Table 4 shows the clean and attack performance when T is first jointly trained with a PreActResnet18 classifier (f); then T is used to generate poisoned samples to attack another classifier f′ whose network can be either similar (i.e., PreActResnet18 as in Same Arch.) or different (i.e., Vgg11  as in Different Arch.) from the architecture of f. In other words, this attack strategy of Marksman assumes that the adversary has only access to the training data but does not control the training process or knows the internal structure of f. Our results showed that Marksman is still very effective in attacking these models that are different from the model learned the trigger function. The ASR is almost 100%.
> Please note the difference in the clean accuracy between the Same Arch. and Different Arch. is due to the original performance of the two different architectures, i.e., PreActResnet18 and Vgg11, which is not impacted by our attack. If we compare the performance before and after the attack on Vgg11, the clean performance is almost the same. We will update this results with the performance drop in this table in a later version.
>
> **Q3: The positioning of the paper, in terms of the threat model considered, with respect to other papers in the related work can be improved.**
>
> Thanks for the suggestions. As we discussed in Section 3, we consider the same threat model with prior works on input-aware backdoor attacks [8,28,9]. We study the same scenario as these works where the adversary has full access to the model during the backdoor injection, which is different from the backdoor attacks that only target poisoned data generation. We will expand this section and highlight the differences among various threat models in a later version.
>
> **Q4: Comment in Limitations**
>
> Thanks for the valuable suggestions! We will reorganize the paper and move the comparisons with ReFoolMT and WaNetMT to the main paper, which may also help address the concerns of other reviewers. As also pointed out by other reviewers, we have conducted additional experiments against more recent defenses, NAD and ANP (the references are in the comment of Reviewer a9D8). The results are shown below. We follow the same experimental settings of NAD and ANP, which also assumes a small subset of clean data (5%). The poisoned models crafted by Marksman undergo the blackdoor-erasing process of NAD and ANP and are then evaluated on both clean and backdoor samples. Similarly, we reported the ASR and Accuracy (ACC) of the processed models. As we can observe, for NAD, while ASR decreases, ACC performance degrades even more significantly. This demonstrates that NAD’s defense is ineffective against Marksman’s attack. For ANP, the defensive process does not significantly degrade the ASRs of Marksman, while the ACCs also drop more significantly. Again, ANP is not effective against Marksman’s attack. Similar to our discussion in the main paper for other defenses, we conjecture that the ability to bypass several existing backdoor defenses is due to the fact that Marksman’s attack poison the model in a very different way than the previously studied backdoor attacks. This calls for more defensive studies to counter the more powerful yet sophisticated multi-trigger and multi-payload attacks of Marksman.
>
> Performance Against NAD Defense
>
> | Dataset | ASR    | ACC    |
> |-------- | ------ | -------|
> | CIFAR10 | 0.3654 | 0.0695 |
> | GTSRB   | 0.1144 | 0.0220 |
>
> Performance Against ANP Defense
>
> | Dataset | ASR    | ACC    |
> |-------- | ------ | -------|
> | CIFAR10 | 0.9385 | 0.6474 |
> | GTSRB   | 0.8901 | 0.6512 |

---

> ### Author Response · Authors · 2022-08-07
> **Happy to answer any additional questions/comments**
>
> Thank you again for your valuable comments. If you have any additional questions/comments, please kindly let us know, and we will try to answer them accordingly.

---

### Meta-Review · Area_Chair_Diud · 2022-08-27

**Recommendation:** Accept
**Confidence:** Less certain

**Metareview:**

This paper introduces a backdoor attack that allows an attacker to cause an input to become any target class, as opposed to just one target class. The reviewers mostly liked this paper and found the attack interesting and useful. The main weakness was in the comparison to prior work, but here the authors have addressed much of this in the comments (and I hope they will make the necessary adjustments in the final version of the paper). The paper also has a limited evaluation against backdoor defenses, but given the goal of this attack is to show a new attack technique and not evade existing defenses I believe it is okay to push this to future work.

**Award:**

No

---

### Decision · Program_Chairs · 2022-09-14

Accept